# Perioperative Antibiotic Prophylaxis: Knowledge and Attitudes among Resident Physicians in Italy

**DOI:** 10.3390/antibiotics9060357

**Published:** 2020-06-25

**Authors:** Concetta Paola Pelullo, Angela Pepe, Francesco Napolitano, Nicola Coppola, Gabriella Di Giuseppe

**Affiliations:** 1Department of Experimental Medicine, University of Campania “Luigi Vanvitelli”, 80135 Naples, Italy; concettapaola.pelullo@unicampania.it (C.P.P.); angela.pepe.84@virgilio.it (A.P.); francesco.napolitano2@unicampania.it (F.N.); 2Department of Mental Health and Public Medicine, Section of Infectious Diseases, University of Campania “Luigi Vanvitelli”, 80131 Naples, Italy; nicola.coppola@unicampania.it

**Keywords:** antibiotics, attitude, case vignettes, Italy, knowledge, perioperative antibiotic prophylaxis, resident physicians, survey

## Abstract

The aim of this study is to evaluate knowledge and attitudes on the perioperative antibiotic prophylaxis (PAP) among surgery and anesthesiology resident physicians in Italy. A Web-based national survey of Italian surgery and anesthesiology resident physicians was conducted between March 2018 and January 2019. Participants completed a questionnaire and three case vignettes for each specialty. Of the 1282 resident physicians selected, 466 completed the online questionnaire for a response rate of 36.3%. More than half of the sample were female (52.9%), and the mean age was 30 years. A total of 36.3% of the participants had an adequate knowledge score about PAP. Multiple logistic regression analysis showed that resident physicians in general surgery compared to those in anesthesiology, those who agreed that PAP must be performed within 60 min before surgical incision, and those who were aware regarding the availability about the availability of national guidelines on PAP, were significantly more likely to have adequate knowledge about PAP. Moreover, 14% of participants were very concerned that patients may contract surgical site infections during hospitalization. These findings should be useful to promote educational intervention specifically targeted for surgery and anesthesiology resident physicians organizing training course on PAP, to improve the correct antibiotic use and to prevent healthcare-associated infections.

## 1. Introduction

Surgical site infections (SSIs) are one of the main healthcare-associated infections (HAIs) and represent the most common complication of the patients who undergo surgical procedures with significant consequences on patients’ health and healthcare services due to the morbidity and related mortality, longer length of hospital stay, readmissions, and increased healthcare expenditure [1]. Indeed, in Europe, SSIs are the second most frequent cause of HAIs and occurs from 0.6% to 9.5% of patients depending by type of surgical procedures [2]. Moreover, in Italy, a prevalence of SSIs has been observed in 2.6% of surgical procedures in 2009–2011 [3].

It is well established that SSIs depend on many factors related to the patients’ and surgical procedures’ characteristics, including the presence of chronic conditions, type and duration of surgery, and perioperative antibiotic prophylaxis (PAP). In particular, the antibiotic management before, during, and after surgical procedures is a crucial moment [4] because appropriate antibiotic use requires an accurate choice of type of antibiotic, timing of administration, and duration of prophylaxis in perioperative period [5,6]. However, despite the availability of several guidelines developed to allow an appropriate use, the adherence to correct practices regarding the use of antibiotic in surgery is worryingly low and antibiotics are used excessively and inappropriately for the prevention of SSIs with negative consequences on the quality of care, such as side effects, onset of antimicrobial resistance, and increased care costs [7].

Surgeons and anesthesiologists are the healthcare workers responsible for prescribing the perioperative antibiotic prophylaxis, and therefore the correct education and training about this topic is crucial in these groups. In Italy, the medical education system consists of different residency programs in clinical, surgical and non-clinical sectors. During their training course, resident physicians participate in the healthcare activities with the supervision of a tutor. At the end of the training course, the specialization diploma is issued in the specific sector and then they can be selected into public or accredited facilities of the national healthcare service [8]. Therefore, it seemed interesting to perform a survey in order to investigate the knowledge and attitudes of the resident physicians in surgery and anesthesiology wards regarding the PAP. The results could provide relevant information to the directors responsible of their training in order to plan effective educational intervention on this issue and thus to improve the appropriateness of antibiotic use and the quality of surgical care.

In literature, several studies have been conducted about HAIs [9,10,11,12,13], whereas very few investigations have focused on the knowledge, attitudes, and practice of physicians about the PAP [14,15,16,17] and no study involved resident physicians. Therefore, this survey had two primary objectives. The first was to evaluate the knowledge and attitudes among a large sample of surgery and anesthesiology resident physicians in Italy on the PAP and the second was to identify the determinants of these outcomes of interest.

## 2. Results

Of the 1282 resident physicians selected, a total of 466 completed the online questionnaire for a response rate of 36.3%. Table 1 summarizes the personal and professional characteristics of the participants. More than half of the sample was female (52.9%) and it is consistent with the gender ratio of physicians in Italy, the mean age was 30 years, and mean length since their degree was four years. One-third of participants (33%) were resident physicians in anesthesiology, 21.7% in general surgery, and 45.3% in surgical specialties, one fourth were enrolled in the first year of training. Moreover, the median number of surgical procedures which the physicians had witnessed in the last twelve months was 100 (interquartile range = 60–200).

The survey responses related to knowledge about PAP are reported in Table 2. Approximately one-third (37.8%) of resident physicians were aware that in their hospital there was an Infection Control Committee, only 13.7% knew the Infection Risk Index, and 24% were aware regarding the availability of national guidelines on PAP in their hospital. More than half of sample (53.5%) declared that recommendations provided by the guidelines on PAP had been always followed in the surgical operations performed during their medical training and only 7.3% indicated that this is never done. Moreover, 86.7% of physicians indicated that the type of surgery that requires PAP was the clean-contaminated surgery when indicated in association with other class of surgery, 4.5% only this class in according to national guideline and 24% believed that PAP is used also for clean surgery in association with or without other type of surgery.

Table 3 shows the resident physicians’ knowledge for each vignette about type of antibiotic, timing of first dose and total length of prophylaxis by specialty. The average of global vignette score was 4.7 ± 2 (0–9).

Moreover, the 36.3% of the participants had adequate knowledge score about type of antibiotic used, the timing of its administration, and the length of the prophylaxis in the case vignettes. Multiple logistic regression analysis showed that resident physicians in general surgery compared to those in anesthesiology (OR = 2.65; 95% CI = 1.66–4.21), those who agreed that PAP must be performed within 60 min before surgical incision (OR = 1.82; 95% CI = 1.04–3.17), those who were aware regarding the availability about the availability of national guidelines on PAP (OR = 1.69; 95% CI = 1.07–2.67), were significantly more likely to have adequate knowledge about type of antibiotic used, the timing of its administration, and the length of the prophylaxis in the case vignettes (Table 4).

Regarding the respondents’ attitude towards PAP, a large majority (81.9%) agreed that PAP must be performed within 60 min before surgical incision and almost all (97%) agreed that the inappropriate choice of the antibiotic may cause antimicrobial resistance. Moreover, one in five of participants (18.3%) agreed with the statement that PAP should be continued beyond 24 h after surgery and only 10.3% do not agreed that several surgical procedures do not require the antibiotic administration.

In regards to perceptions about SSIs, more than two-thirds (68%) agreed that SSIs are preventable infections, and a large majority (79.8%) were concerned that patients may contract SSIs during hospitalization with a mean score of 7.1 out of a maximum score of 10. A multivariate logistic regression model was built to investigate regarding the factors associated with the resident physicians who were very concerned that patients may contract SSIs during hospitalization (14%). The analysis results showed that those who were aware that in their hospital there was an Infection Control Committee (OR = 3.36; 95% CI = 1.85–6.1), were more likely to be very concerned that patients may contract SSIs during hospitalization. Moreover, participants who were resident physicians in general surgery (OR = 0.1; 95% CI = 0.03–0.36) and in surgical specialties (OR = 0.17; 95% CI = 0.06–0.48) compared to those in anesthesiology were less concerned that patients may contract SSIs during hospitalization. Participants were asked to indicate the reasons for which they were concerned that patients could have SSIs and the most frequent reasons reported were that SSIs delay patient healing (88.7%), that SSIs lead to medical-legal disputes (23.2%), and the SSIs’ high rate of incidence (18%). Instead, the most frequent reasons reported by those who were not concerned were that in their surgical ward there are greatly attention to the prevention of SSIs (56%), that SSIs rarely occur (48.7%), and that are easily to treat (12%).

Almost half of the participants (47%) were very concerned about the development of multi-resistant antibiotic bacteria. The results of multivariate logistic regression analysis showed that those who knew the Infection Risk Index (OR = 2.29; 95% CI = 1.28–4.1), were more likely to be very concerned about the development of multi-resistant antibiotic bacteria, whereas resident physicians in general surgery (OR = 0.51; 95% CI = 0.27–0.98) and in surgical specialties (OR = 0.54; 95% CI = 0.31–0.94) compared to those in anesthesiology were less concerned.

Moreover, 34.3% considered useful PAP in order to reduce the incidence of SSIs with an overall mean value of 8.3, out of a maximum score of 10. The results of multivariate logistic regression analysis showed that physicians who had adequate knowledge score about PAP (OR = 1.65; 95% CI = 1.07–2.55), those who were aware that in their hospital there was an Infection Control Committee (OR = 1.81; 95% CI = 1.18–2.78), those who were aware that SSIs are preventable infections (OR = 2.02; 95% CI = 1.17–3.48), were more likely to consider useful PAP in order to reduce the incidence of SSIs. Moreover, resident physicians in general surgery (OR = 0.39; 95% CI = 0.22–0.72), compared with those in anesthesiology were less likely to consider useful PAP in order to reduce the incidence of SSIs.

Overall, 86.4% reported having received information about PAP and the major sources by which the resident physicians obtain information were in order PAP guidelines (86.5%), colleagues (29.9%), and books (29.8%). Moreover, one-third (32.5%) had attended training courses on PAP in the last year and almost all (94.5%) reported that they felt the need to receive additional information about PAP.

## 3. Discussion

Previous studies published worldwide have evaluated knowledge, attitudes and practice about PAP among physicians, but to the best of our knowledge, the present study is the first to describe the pattern of PAP among resident physicians in Italy and, in particular, using vignettes to evaluate knowledge about this topic. Therefore, the comparison of our results with those of other studies is very difficult because of different samples, objectives and methodologies used.

Our results, overall, showed a deficiency in knowledge about PAP, while it indicated satisfactory attitudes towards SSIs as preventable infections and that an inappropriate choice of the antibiotic may cause antimicrobial resistance. In this current study, it was found that only 36.3% of resident physicians had an adequate level of knowledge about the type of antibiotic used, the timing of its administration, and the length of the prophylaxis in that were illustrated in the case vignettes. Moreover, few knew the Infection Index Risk and only 24% were aware of the availability of national guidelines on PAP in their hospital. This is an interesting finding that denotes the need for improvement of quality of care through patient safety interventions.

The knowledge of resident physicians varied among the different specialties. Indeed, the knowledge was adequate among general surgery compared with those in anesthesiology. This result could be explained because, usually in Italy, the administration of PAP is under the responsibility of surgeons. This is contrary of that found in another study conducted in France, in which anesthesiologists had better knowledge compared to surgeons [17].

A total of 86.7% of respondents individuated that the type of surgery that require PAP was the clean-contaminated surgery when indicated in association with other class of surgery, while 4.5% correctly identified only this class in according to national guideline. Indeed, 24% believed that PAP is used also for clean surgeries in association with or without other type of surgery. This result is less pronounced than another study conducted in the United Kingdom, in which 73% of surgeons gave antibiotics in clean surgeries [18].

With regards to attitudes of resident physicians, the vast majority believed that PAP was useful to reduce the incidence of SSIs, but our results reveal that most of the three quarter of respondents were concerned that patients may contract SSIs during hospitalization. Therefore, although in our study there was an inadequate level of knowledge, there was also a high perception of risk, whereby educational interventions are absolutely necessary. Similar results are showed in other studies, although they are conducted on other populations. Indeed, in a study conducted in Italy among the general population, 79.8% of patients were worried about contracting hospital-associated infections [19]. These findings suggest the importance of training and formation for all healthcare workers, in particular for surgeons and anesthesiologists, who are personally interested in the administration of PAP, in order to prevent SSIs and to correctly inform their patients about these issues.

Respondents were also asked about their sources of information about PAP. The majority (86.5%) received information on guidelines, and about 30% from colleagues and books. Moreover, only one-third had attended training courses on PAP. Therefore, these results highlighted the importance of educational interventions, which are crucial in providing information about PAP among physicians, that could improve their knowledge on this issue. This is in accordance with another published study by some of the authors in this study [20,21,22,23,24,25], in which searching information could be useful to improve the level of knowledge of healthcare workers.

This study had certain limitations that must be noted in the interpretation of the results. First, this was a cross-sectional study and it is difficult to demonstrate a temporal relationship between explanatory variables and the different outcomes of interest. Second, by using a questionnaire with self-reported information, it may be subject to desirability biases, because the participants may have responded to questions regarding their practice in a socially desirable way, even though questionnaire was anonymous. Third, this is an online survey and with a low response rate, but certainly higher than other studies with similar methodology [26,27]. This study also has important strengths. This is the first study that investigates the knowledge and attitudes of resident physicians towards PAP using a nationally representative Italian sample and, therefore, provides important information regarding this population previously not investigated.

## 4. Materials and Methods 

A Web-based national survey of Italian surgery and anesthesiology resident physicians was performed between March 2018 and January 2019. Participants were selected using a two-stage cluster sampling. In the first stage, from the list of Italian public Universities, 15 of them were randomly selected using a computer-generated list of random numbers. In the second stage a random sample of 44 University-based Medical Schools of Surgery (General, Cardiac, Thoracic, Plastic, Vascular, Orthopedic, Gynecology, Urology, Otolaryngology, and Ophthalmology) and Anesthesiology was selected and to all resident physicians who attended the Medical Schools were sent a questionnaire by e-mail.

The sample size was calculated based on the estimation that 50% of the resident physicians were aware about the appropriate preoperative antibiotic prophylaxis, a 95% confidence interval, a response rate of 80%, and an error of 0.05. Therefore, the minimum number of participants required was estimated at 480.

### 4.1. Procedure

Prior to the start of the survey, approval to perform the study was requested to the Directors of the selected Medical Schools through a letter that explained the aims and the methodology of the investigation and data collection. Following the approval, the Directors sent to the research team the email-addresses of potential participants with the resident physicians’ consent. The anonymous questionnaires were distributed via e-mail using the platform Google Drive (Google Inc. Mountain View, CA, USA) and three repeat requests were sent to non-responders in order to improve the response rate. The e-mail contained a cover letter that explained the purposes of the study and the methods of data collection, assured that the survey was voluntary and that all data were collected and analyzed anonymously, and specified that participants who send back the questionnaire gave their informed consent to participate. No monetary compensation or gift was given to the respondents. Data collection was finalized on January 31, 2019.

### 4.2. Survey Instrument

Prior to the start of the study, the questionnaire was pre-tested on 50 resident physicians, not included in the final sample, to ensure the clarity and the validity of the questionnaire.

The questionnaire was composed of the following four sections: 1) personal and professional characteristics (age, gender, year of graduation, specialty, geographic area of activity, Medical Schools attended, number of surgical procedures witnessed in the last year, number of SSIs in their ward in the last year, whether or not there are the Committee for the of Hospital Infections Control and perioperative antibiotic prophylaxis guidelines in their work place, and whether or not antibiotic prophylaxis was required in surgery classified as clean, clean-contaminated, contaminated, and dirty-contaminated); 2) knowledge about PAP; 3) attitude towards PAP (usefulness of PAP in order to reduce the incidence of SSIs, awareness that SSIs are preventable infections, awareness that PAP must be performed within 60 min before surgical incision); 4) source and need of additional information about PAP.

The study protocol as well as the questionnaire to collect the data were approved by the Ethics Committee of the Teaching Hospital of the University of Campania “Luigi Vanvitelli” (approval number 37/2018).

### 4.3. Case Vignettes

Participants’ knowledge about the PAP was evaluated through case vignettes. A vignette represents a special type of teaching case based on case history of a patient, used to measure physician’s knowledge [14]. In particular, a set of three case vignettes for each surgical specialty was created representing surgical procedures performed on patients suffered of certain conditions and with several risk factors for SSIs. In each vignette, the age of patient, some clinical characteristics and the expected length of the intervention was included. Moreover, for each surgical procedure, participants were asked to indicate the type of antibiotic, the timing of its administration, and the total length of the prophylaxis as a single or multiple doses of antibiotic administered within 24 h. Response options included a list of choices, with a only one correct response (S1: Example of case vignette). The research team wrote the case vignettes. Appropriateness of surgical antibiotic prophylaxis indicated by resident physicians was assessed based on the Italian national guidelines [28].

### 4.4. Outcomes of Interest

To determine the level of knowledge about the three case vignettes, for each surgical procedure, we evaluated the type of antibiotic used, the timing of its administration, and the length of the prophylaxis. Then, we assigned a score of “1” for the correct answer and “0” for the incorrect. Therefore, the total knowledge score was calculated and it ranged from “0” to “9”. This knowledge score was then categorized according to the median of knowledge score into adequate knowledge (score >5) and inadequate knowledge (≤5). In this study, there were four outcomes of interest: (a) adequate knowledge about type of antibiotic used, the timing of its administration, and the length of the prophylaxis in the case vignettes (no = 0; yes = 1), (Model 1); (b) resident physicians who were very concerned that patients may contract SSIs during hospitalization (no = 0; yes = 1), (Model 2); (c) resident physicians who were very concerned about the development of multi-resistant antibiotic bacteria (no = 0; yes = 1), (Model 3); (d) utility of PAP in reducing SSIs (no = 0; yes = 1), (Model 4).

### 4.5. Statistical Analysis

In all the models, the independent variables included were: age (26–30 years = 0; >30 years = 1), gender (male = 0; female = 1), specialty (anesthesiology = 1; general surgery = 2; surgical specialties = 3), geographic area of activity (Southern Italy = 1; Center of Italy = 2; Northern Italy = 3), years of training (other = 0; first = 1), number of surgical procedures witnessed (continuous), SSIs encountered during their activity (no = 0; yes = 1), awareness about Infection Control Committee in their hospital (no = 0; yes = 1), awareness about the availability of national guidelines on PAP (no = 0; yes = 1), type of surgery that require PAP (other type of surgery = 0; clean-contaminated surgery = 1), Infection Index Risk (other = 0; intervention class, American Society of Anesthesiologists (ASA) Score, duration of intervention = 1), awareness that SSIs are preventable infections (no = 0; yes = 1), awareness that PAP must be performed within 60 min before surgical incision (no = 0; yes = 1), training courses on PAP (no = 0; yes = 1), sources of information (colleagues/courses/scientific journal/books = 0; PAP guidelines = 1), need of additional information (no = 0; yes = 1). The following variables were also included: adequate knowledge about type of antibiotic used, the timing of its administration, and the length of the prophylaxis in the case vignettes (no = 0; yes = 1), in Models 2, 3 and 4.

Data analyses were performed using Stata statistical software, version 15 (StataCorp LLC, College Station, TX, USA) [29]. T-tests and chi-square tests were conducted to assess the univariate associations between each of the independent characteristics and the different outcomes of interest. After performing the bivariate analyses, those variables found to be associated with the outcomes of interest at the *p*-value ≤0.25 level were subsequently introduced into multivariate regression models. Then, the associations between the independent variables and the dichotomous outcomes of interest were performed with stepwise logistic regression analysis. Variables were selected for the multivariate models using a *p*-value of 0.2 for entry and 0.4 for exclusion. Odds ratios (ORs) and 95% confidence intervals (CIs) are presented for logistic regression models and *p*-values ≤0.05 were considered to be statistically significant.

## 5. Conclusions

Since surgeons and anesthesiologists play a crucial role in the administration of antibiotics, the implementation of PAP interventions, already during medical schools, that aim to increase their knowledge are of the utmost importance in preventing infection in their patients and in limiting the economic impact of infection in healthcare. In conclusion, PAP in medical resident training, knowledge and adherence to hospital PAP protocol should change the antibiotic use in surgical prophylaxis and improve knowledge and practice of PAP among this population.

## Figures and Tables

**Table 1 antibiotics-09-00357-t001:** Socio-demographic and professional characteristics of the study population.

	All Specialties (N = 466)		Anesthesiology (N = 154)		General Surgery (N = 101)		Surgical Specialties (N = 211)	
	N	%	N	%	N	%	N	%
Age, years	30.4 ± 2.86 (26–50) *		31 ± 3.69 (26–50) *		29.8± 2.04 (27–36) *		30.2 ± 2.34 (27–40) *	
Gender								
Male	212	47.1	65	42.2	52	51.5	95	48.7
Female	238	52.9	89	57.8	49	48.5	100	51.3
Geographic area of activity								
Southern Italy	211	45.3	140	90.9	22	21.8	49	23.2
Center of Italy	53	11.4	14	9.1	7	6.9	32	15.2
Northern Italy	203	43.3	-	-	72	71.3	130	61.6
Number of years since degree	4 ± 1.84 (1–12) *		3.9 ± 2.06 (1–12) *		3.8 ± 1.67 (1–7) *		4.2 ± 1.74 (1–12) *	
Years of training								
First year	111	24.5	53	34.9	24	23.8	34	17
Second to sixth year	342	75.5	99	65.1	77	76.2	166	83
Number of surgical procedures witnessed	100 (60–200) ^+^		100 (50–200) ^+^		110 (80–200) ^+^		120 (80–204) ^+^	

* Mean ± Standard deviation (Range). ^+^ Median (interquartile range). Numbers for each item may not add up to total number of study population due to missing values.

**Table 2 antibiotics-09-00357-t002:** Respondents’ knowledge and sources of information about perioperative antibiotic prophylaxis.

	*N*	%
Type of surgery that require PAP		
Clean-contaminated surgery	21	4.5
Other type of surgery	445	95.5
Infection Index Risk		
Intervention class, ASA, duration of intervention	63	13.7
Other	398	86.3
Infection Control Committee in their hospital		
Yes	176	37.8
No	290	62.2
Availability of national guidelines on PAP in their hospital		
Yes	112	24
No	354	76
Sources of information about PAP		
PAP guidelines/Colleagues/Books	60	15.3
National guidelines on PAP	332	84.7
Training courses on PAP		
Yes	149	32.5
No	309	67.5

Numbers for each item may not add up to total number of study population due to missing values.

**Table 3 antibiotics-09-00357-t003:** Respondents’ knowledge for each vignette about type of antibiotic, timing of first dose and total duration by specialty.

	Anesthesiology			General Surgery			Surgical Specialties		
	Vignettes			Vignettes			Vignettes		
	1	2	3	1	2	3	1	2	3
	n (%)	n (%)	n (%)	n (%)	n (%)	n (%)	n (%)	n (%)	n (%)
**Type of antibiotic**									
Incorrect	131 (86.7)	145 (99.3)	36 (24.5)	12 (12.1)	6 (6)	72 (72)	46 (21.8)	128 (72.3)	81 (44)
Correct	20 (13.3)	1 (0.7)	111 (75.5)	87 (87.9)	94 (94)	28 (28)	165 (78.2)	49 (27.7)	103 (56)
**Timing of first dose**									
Incorrect	18 (12.7)	19 (13.9)	21 (15.2)	34 (36.6)	37 (37.8)	30 (36.6)	83 (44.4)	75 (43.3)	53 (40.8)
Correct	124 (87.3)	118 (86.1)	117 (84.8)	59 (63.4)	61 (62.2)	52 (63.4)	104 (55.6)	98 (56.7)	77 (59.2)
**Total length of the prophylaxis**									
Incorrect	42 (30.7)	53 (49.6)	70 (50.4)	17 (18.5)	37 (38.1)	21 (25.9)	44 (23.9)	79 (45.4)	60 (44.8)
Correct	95 (69.3)	54 (50.4)	69 (49.6)	75 (81.5)	60 (61.9)	60 (74.1)	140 (76.1)	95 (54.6)	74 (55.2)

Numbers for each item may not add up to total number of study population due to missing values.

**Table 4 antibiotics-09-00357-t004:** Multivariate logistic analyses to characterize factors associated with the different outcome of interest.

Variable	OR ^+^	SE ^°^	95% CI ^^^	*p*-Value
Model 1. Adequate knowledge about type of antibiotic used, the timing of its administration, and the length of the prophylaxis in the case vignettes				
Log likelihood = −276.42, χ^2^ = 36.6 (6 df), *p* < 0.0001 (sample size = 449)				
Specialty				
Anesthesiology *	1 *			
General surgery	2.65	0.63	1.66–4.21	<0.001
Who were aware about the availability of national guidelines on PAP	1.69	0.39	1.07–2.67	0.023
Who agreed that PAP must be performed within 60 min before surgical incision	1.82	0.52	1.04–3.17	0.035
First year of training	1.55	0.36	0.98–2.45	0.059
Center of Italy as geographic area of activity	0.56	0.19	0.28–1.1	0.094
Knowledge of Infection Index Risk	1.45	0.42	0.81–2.57	0.206
Model 2. Resident physicians who were very concerned that patients may contract SSIs during hospitalization				
Log likelihood = −153.56, χ^2^ = 50.4 (8 df), *p* < 0.0001 (sample size = 439)				
Specialty				
Anesthesiology *	1 *			
General surgery	0.1	0.07	0.03–0.36	<0.001
Surgical specialties	0.17	0.09	0.06–0.48	0.001
Who were aware about Infection Control Committee in their hospital	3.36	1.02	1.85–6.1	<0.001
Adequate knowledge about type of antibiotic used, the timing of its administration, and the length of the prophylaxis in the case vignettes	1.75	0.53	0.96–3.17	0.068
Female	1.61	0.49	0.88–2.93	0.12
Northern Italy as geographic area of activity	2.04	1.09	0.72–5.81	0.181
Age	1.33	0.4	0.74–2.41	0.338
Who were aware that SSIs are preventable infections	1.44	0.58	0.66–3.19	0.36
Model 3. Resident physicians who were very concerned about the development of multi-resistant antibiotic bacteria				
Log likelihood = −287.32, χ^2^ = 32.3 (6 df), *p* < 0.0001 (sample size = 439)				
Knowledge of Infection Index Risk	2.29	0.68	1.28–4.1	0.005
Specialty				
Anesthesiology *	1 *			
General surgery	0.51	0.17	0.27–0.98	0.044
Surgical specialties	0.54	0.15	0.31–0.94	0.03
Center of Italy as geographic area of activity	0.56	0.19	0.28–1.09	0.086
Northern Italy as geographic area of activity	0.64	0.18	0.37–1.12	0.118
Age	1.19	0.24	0.8–1.78	0.397
Model 4. Utility of PAP in reducing SSIs				
Log likelihood = −267.89, χ^2^= 37.3(8 df), *p* < 0.0001 (sample size = 447)				
Specialty				
Anesthesiology *	1 *			
General surgery	0.39	0.12	0.22–0.72	0.002
Surgical specialties	0.82	0.19	0.52–1.3	0.397
Who were aware about Infection Control Committee in their hospital	1.81	0.4	1.18–2.78	0.006
Who were aware that SSIs are preventable infections	2.02	0.56	1.17–3.48	0.011
Adequate knowledge about type of antibiotic used, the timing of its administration, and the length of the prophylaxis in the case vignettes	1.65	0.36	1.07–2.55	0.023
Who agreed that PAP must be performed within 60 min before surgical incision	1.73	0.51	0.96–3.1	0.066
Female	1.45	0.31	0.96–2.2	0.076
Knowledge of Infection Index Risk	1.58	0.47	0.89–2.82	0.119

**^+^** Odds Ratio, **^°^** Standard Error, **^^^** Confidence Interval, * Reference category.

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
