# Peer review of "Perioperative Antibiotic Prophylaxis: Knowledge and Attitudes among Resident Physicians in Italy"

_antibiotics, 2020, doi:10.3390/antibiotics9060357_

Round 1

Reviewer 1 Report

Thank you for giving me the opportunity to review the article. The authors conducted a study on the knowledge and attitudes on the perioperative antibiotic prophylaxis among in Italy. The topic is important to improve medical education, but several methodological concerns exist. I listed the comments below.

Comments:

Abstract:

  1. The authors should add the study period.
  2. The authors should add the number of potential participants and physicians who participated in this study (with response rate) should be added.
  3. I understood that the findings obtained from this study should be useful to promote educational intervention, but the authors should conclude the abstract with a more detailed statement.

Results:

  1. More than the half of the sample was female. Is it consistent with the gender ratio of physicians?
  2. The number of surgical procedures witnessed should be expressed as median and IQR.
  3. Only 24% of participants answered “Yes” to the availability of national guidelines on PAP in their hospital, but 84.7% used national guidelines on PAP as a source of information about PAP. Why this gap can be occurred? The reviewer thought that most of participants referred guideline through the internet.

Discussion:

  1. The authors should discuss about the low response rate (approx. 35%) of this study.
  2. The authors should discuss about the medical education system in Italy. Potential readers in foreign countries cannot understood the background of this study well.

Materials and Methods:

  1. How did the authors randomly select universities participated in this study?
  2. The authors should add the details of the vignette used in this study.
  3. How to evaluate the knowledge for each vignette? The authors should provide the examination used in this study. These Yes/No answer provided in the Table 3 cannot evaluate the knowledge.

Author Response

Abstract

  1. (Lines 15-16) As suggested, in the Abstract section we have added the study period.
  2. (Lines 16-17) As suggested, in the Abstract section we have added the number of participants and the response rate.
  3. (Lines 27-28) As suggested, in the Abstract section we have better specified the educational intervention to promote. Therefore, we have modified the Abstract because it should be a total of about 200 words maximum.

Results

  1. In response to the point regarding the fact that more than the half of the sample was female, it is consistent with the gender ratio of physicians in Italy. Indeed, the last data of the Italian National Federation of Physicians showed that a large majority of physicians aged <50 years were female [https://portale.fnomceo.it/8-marzo-il-sorpasso-delle-donne-medico-sotto-i-65-anni-sono-piu-degli-uomini/].
  2. (Lines 78-79) As suggested, in Results section and in Table 1 we have expressed the surgical procedures witnessed as median and IQE.
  3. In response to the point regarding national guidelines about PAP, we specify that 84.7% of medical resident knew the availability of national guidelines and therefore they used them as source of information about PAP, but only 24% affirmed that were really used in their hospital.

Discussion

  1. In response to the point regarding the low response rate, as reported in recent investigations (McLeod et al., 2013; Cunningham et al., 2015), the response rate for web-surveys is lower compared to other strategies. Moreover, low response rate has been reported also in online surveys carried out among healthcare workers (Penna et al., 2013) and medical resident (Costantino et al., 2014).
  2. In response to the point regarding the medical education system in Italy, in the Introduction section we have specified the medical education in Italy and we have included appropriate reference.

Materials and Methods

  1. As specified in Materials and Methods section, from the list of the Italian public Universities, 15 were randomly selected. Subsequently, a random sample of 44 University-based Medical Schools of Surgery (General, Cardiac, Thoracic, Plastic, Vascular, Orthopedic, Gynecology, Urology, Otolaryngology, and Ophthalmology) and Anesthesiology have been selected.
  2. (Lines 255-256, 259) As suggested, we have better specified the detail of vignettes used in this study.
  3. In response to the point regarding knowledge, we specify that it was created a set of 3 case vignettes for each surgical specialty representing surgical procedures performed on patients suffered of certain conditions and with several risk factors for SSIs. Moreover, for each surgical procedure, participants were asked to indicate the type of antibiotic, the timing of its administration, and the total length of the prophylaxis as a single or multiple doses of antibiotic administered within 24 hours. Response options included a list of choices, with a only one correct response. In Table 3 we have only dichotomized correct versus incorrect responses.

    Reviewer 1

    Abstract

    1. (Lines 15-16) As suggested, in the Abstract section we have added the study period.
    2. (Lines 16-17) As suggested, in the Abstract section we have added the number of participants and the response rate.
    3. (Lines 27-28) As suggested, in the Abstract section we have better specified the educational intervention to promote. Therefore, we have modified the Abstract because it should be a total of about 200 words maximum.

    Results

    1. In response to the point regarding the fact that more than the half of the sample was female, it is consistent with the gender ratio of physicians in Italy. Indeed, the last data of the Italian National Federation of Physicians showed that a large majority of physicians aged <50 years were female [https://portale.fnomceo.it/8-marzo-il-sorpasso-delle-donne-medico-sotto-i-65-anni-sono-piu-degli-uomini/].
    2. (Lines 78-79) As suggested, in Results section and in Table 1 we have expressed the surgical procedures witnessed as median and IQE.
    3. In response to the point regarding national guidelines about PAP, we specify that 84.7% of medical resident knew the availability of national guidelines and therefore they used them as source of information about PAP, but only 24% affirmed that were really used in their hospital.

    Discussion

    1. In response to the point regarding the low response rate, as reported in recent investigations (McLeod et al., 2013; Cunningham et al., 2015), the response rate for web-surveys is lower compared to other strategies. Moreover, low response rate has been reported also in online surveys carried out among healthcare workers (Penna et al., 2013) and medical resident (Costantino et al., 2014).
    2. In response to the point regarding the medical education system in Italy, in the Introduction section we have specified the medical education in Italy and we have included appropriate reference.

    Materials and Methods

    1. As specified in Materials and Methods section, from the list of the Italian public Universities, 15 were randomly selected. Subsequently, a random sample of 44 University-based Medical Schools of Surgery (General, Cardiac, Thoracic, Plastic, Vascular, Orthopedic, Gynecology, Urology, Otolaryngology, and Ophthalmology) and Anesthesiology have been selected.
    2. (Lines 255-256, 259) As suggested, we have better specified the detail of vignettes used in this study.
    3. In response to the point regarding knowledge, we specify that it was created a set of 3 case vignettes for each surgical specialty representing surgical procedures performed on patients suffered of certain conditions and with several risk factors for SSIs. Moreover, for each surgical procedure, participants were asked to indicate the type of antibiotic, the timing of its administration, and the total length of the prophylaxis as a single or multiple doses of antibiotic administered within 24 hours. Response options included a list of choices, with a only one correct response. In Table 3 we have only dichotomized correct versus incorrect responses.

Reviewer 2 Report

This article, described the current state of PAP awareness among resident doctors in Italy, is interesting, however, I have some questions and comments to the author.

Q1. Table 1 shows the geographic area of ​​activity of participants, but there is a difference in the regional distribution of surgeon and anesthesiologist. Is this lacking nutrition in this analysis? Does not this difference affect the results of the study?

Q2. In Table 1, the number of surgical procedures witnessed is considered to be quite wide in each specialty. Is there no correlation between number of experiences and the results of this study, that is, the difference in the recognition of PAP?

Q3. I think it is better to explain the case vignettes and models 1-4 in a separate paragraph and item rather than in a statistical analysis. In addition, the content of "3 case vignettes" is not shown, so its validity is unknown.

Q4. The line 164-166 in “Discussion” describes the difference in PAP recognition from UK. What is the reason for the difference between Italy and UK? Is the resident doctor's training system different?

Q5. What do you think is most effective for the recognition and knowledge of PAP, Infection Control Committee, availability of national guidelines or training courses on PAP?

 Q6. In "Materials & Methods" line 232-233, "--- used to measure physicians knowledge.13 In particular, it ---", I think "13" is a typo, so please cut it.

Author Response

Q1. In response to the point regarding the geographic area of activity, there was a misunderstanding. This variable have included in Materials and Methods, but not included in logistic regression analysis. Therefore, we have retake a multiple logistic regression analysis for Model 1, 2 and 3. In Model 4, after performing the bivariate analysis, this variable had a p-value>0.25 and it was not included in the final model. Moreover, we have corrected the misunderstanding.

Q2. In response to the point regarding the number of procedures witnessed, we specify that this variable was included in all regression models. After performing the bivariate analyses, only those variables found to be associated with the outcomes of interest at the p-value≤0.25 level were subsequently introduced into a multivariate regression model. Therefore, this variable was not included in our analysis.

Q3. (Lines 250, 255-256, 259, 265, 277) As suggested, the Materials and Methods section has been sub-divided into further separate headings including: case vignettes, outcome of interest and statistical analysis. Moreover, we have better specified the detail of 3 case vignettes used in this study.

Q4. The training medical education system in UK is different to Italy. After their undergraduate training, physicians undergo 2 years of foundation program training before choosing a specialty in a specific field of medicine to develop the competencies required to practice in their chosen discipline. Despite this different setting, there is probably no specific training on PAP in their courses which could explain the discrepancy in comparison with our results

Q5. Despite the available guidelines, the inappropriate perioperative antibiotic prophylaxis remains a common phenomenon in many countries. It is well known that a multi-disciplinary approach is needed in order to improve the appropriate use of antibiotics and to decrease adverse events, antibiotic resistance, and associated expenditures. Therefore, educational and training interventions are needed aimed at all professionals involved in the use of antibiotics, such as physicians, pharmacists, epidemiologists, and managers of healthcare facilities.

Q6. (Line 253) There was a typo error. “13” was referred to reference number 13, therefore now was reported into square brackets.

Round 2

Reviewer 1 Report

Thank you for giving me the opportunity to review the revised version of this article. The authors revised the manuscript according to the comments mostly. However, I thought that minor additional changes should be needed before acceptance for publication. I listed the comments below.

AR, authors’ response; AC, additional comment

Results:

  1. More than the half of the sample was female. Is it consistent with the gender ratio of physicians?
    AR: In response to the point regarding the fact that more than the half of the sample was female, it is consistent with the gender ratio of physicians in Italy.
    AC: This should be mentioned briefly in the Results or Discussion section.

Materials and Methods:

  1. How did the authors randomly select universities participated in this study?
    AR: As specified in Materials and Methods section, from the list of the Italian public Universities, 15 were randomly selected. Subsequently, a random sample of 44 University-based Medical Schools of Surgery (General, Cardiac, Thoracic, Plastic, Vascular, Orthopedic, Gynecology, Urology, Otolaryngology, and Ophthalmology) and Anesthesiology have been selected.
    AC: I had understood the process. However, I would like to know “how” randomly select the study sites. If possible, the authors should write it in the section.
  2. The authors should add the details of the vignette used in this study.
    AR: (Lines 255-256, 259) As suggested, we have better specified the detail of vignettes used in this study.
    AC: Thank you for writing about the detail. However, the reviewer would like you to upload the vignette used in this study as a supplementary material for the potential readers of this paper.

Author Response

  1. As suggested, in the Results section we have added briefly the gender ratio of physicians in Italy.
  2. As suggested, in the Materials and Methods section we have better specified the selection of universities.
  3. As suggested, we have included an example of case vignette used in this study as Supplementary file.

Reviewer 2 Report

Thank you for correcting the manuscript.

Author Response

My colleagues and I are most grateful for the extremely positive tone of the comments.